# Interplay between topological valley and quantum Hall edge transport

Fabian R. Geisenhof [1], Felix Winterer [1], Anna M. Seiler[1,2], Jakob Lenz[1], Ivar Martin [3] & R. Thomas Weitz [1,2,4,5✉]

An established way of realising topologically protected states in a two-dimensional electron gas is by applying a perpendicular magnetic field thus creating quantum Hall edge channels. In electrostatically gapped bilayer graphene intriguingly, even in the absence of a magnetic field, topologically protected electronic states can emerge at naturally occurring stacking domain walls. While individually both types of topologically protected states have been investigated, their intriguing interplay remains poorly understood. Here, we focus on the interplay between topological domain wall states and quantum Hall edge transport within the eight-fold degenerate zeroth Landau level of high-quality suspended bilayer graphene. We find that the two-terminal conductance remains approximately constant for low magnetic fields throughout the distinct quantum Hall states since the conduction channels are traded between domain wall and device edges. For high magnetic fields, however, we observe evidence of transport suppression at the domain wall, which can be attributed to the emergence of spectral minigaps. This indicates that stacking domain walls potentially do not correspond to a topological domain wall in the order parameter.

[1] Physics of Nanosystems, Department of Physics, Ludwig-Maximilians-Universität München, Munich, Germany. [2] 1st Physical Institute, Faculty of Physics, University of Göttingen, Göttingen, Germany. [3] Materials Science Division, Argonne National Laboratory, Lemont, IL, USA. [4] Center for Nanoscience (CeNS), LMU Munich, Munich, Germany. [5] Munich Center for Quantum Science and Technology (MCQST), Munich, Germany. ✉email: thomas.weitz@uni-goettingen.de

Electrons near the Fermi surface of two-dimensional hexagonal materials typically occupy two or more distinct electronic valleys. The valley index adds to the carrier's charge and spin, enabling additional channels for spontaneous symmetry breaking at low temperatures, whereby valleys are polarised independently or in combination with charge and spin degrees of freedom[1,2]. The most direct way to induce non-trivial valley response is by breaking sublattice symmetry. This occurs naturally in boron nitride, which makes it a quantum valley Hall insulator[3]. In Bernal-stacked bilayer graphene, the same effect is achieved by applying an interlayer bias[4]. Moreover, by spatially varying its sign, topological domain walls can be created, which exhibit one-dimensional (1D) electronic channels with quantised conductance[4], resilient to backscattering[5]. These electronic domain-wall states provide a flexible platform to study 1D transport[6–8] and correlated physics[9–11]. However, creating them by electrostatic gating is technically challenging. Fortunately, similar physics transpire at stacking domain walls (DWs) in bilayer graphene, where the stacking arrangement of graphene layers changes from AB to BA[12]. Such domain walls are common in naturally Bernal-stacked bilayer graphene[13–15] and even ubiquitous in twisted bilayer graphene[16,17], which is known for hosting superconductivity at a certain twist angle[18]. When a uniform electric field is applied to a bilayer graphene flake with a DW, topologically protected valley-helical states emerge along the dislocation, surrounded by insulating bulk[12,14,19]. Critically for the present work, stacking domain walls can have much richer interplay with spontaneous symmetry breaking in bilayer graphene[20–27] compared to artificially created ones, as not being forced by applied bias to have charge imbalance between layers. The interplay between stacking domain walls and spontaneous symmetry breaking is of peculiar interest in the presence of a quantising magnetic field, since bilayer graphene exhibits a very rich phase diagram owing to the eightfold degeneracy of the zero-energy Landau levels[28–30] (coming from two valleys, two orbital Landau level indices, and two spins – neglecting Zeeman splitting). Interactions lift the degeneracy by generating orderings, leading to quantum Hall plateaus at all integer filling fractions between $-4$ and $4$[24,28–32]. This complex and intriguing regime shows a large variety of ways the internal symmetry can break spontaneously in the absence of externally induced layer polarisation. Within this manifold, the valley, sublattice, and layer index are rigidly locked. Since at the stacking domain wall the roles of the layers are exchanged, any ordering that is not a valley singlet is guaranteed to be affected.

In this work, the goal is to study this interplay by means of transport measurements. It cannot be fully explored in the artificial electrostatic domain walls as a matter of principle. We chose freestanding dually gated bilayer graphene devices as an ideal and versatile platform, since on the one side—as indicated by our measurements below—DWs remain stable during processing and suspension, and, on the other side, suspending enables the investigation of quantum transport unaffected by surroundings.

## Results and discussion

### Topological valley transport in the presence of an electric field induced gap.

At first, suitable bilayer graphene flakes were pre-selected using optical microscopy and subsequently investigated with scattering scanning near-field optical microscopy[14,15,33]. Even though flakes show a smooth surface in the topography (Fig. 1a), the corresponding near-field amplitude image (Fig. 1b) can reveal stacking domain walls. Second, contacts were designed in two different configurations, as schematically illustrated in Fig. 1c. Either a DW was contacted on both ends (i.e. the DW goes along the channel separating two distinct domains, one with

AB and one with BA stacking), or, alternatively, no domain wall was within the channel. Two devices are discussed exemplarily in the following: D1-DW of the former and D2 (which has been also investigated in ref. [27].) of the latter type. Data from additional domain-wall containing devices are shown in the Supplementary Information.

Using the dual-gate structure and sweeping the top $V_t$ and bottom gate voltage $V_b$ while tracking the resistance for the two configurations reveals differences in their signatures (Fig. 1d, e). Device D2 (Fig. 1e) shows, consistent with previous measurements, the spontaneously gapped state at the charge neutrality point[20–24] and a phase transition to the insulating fully layer polarised state for increasing electric field[23,24]. The resistance in device D1-DW (Fig. 1d) shows an overall similar behaviour, but with very different values. This becomes more apparent when examining line traces (see Fig. 1f, g). Although the resistance in both devices behaves non-monotonically as a function of increasing $V_t$, which indicates the emergence of the layer antiferromagnetic (LAF) ground state with opposite spins in two layers[1,34,35] at charge neutrality and zero electric field (at $V_t \approx V_b \approx 0$), it remains low in device D1-DW. As discussed below, this is caused by additional charge channels, which mask the insulating phase. Moreover, consistent with previous measurements[7,14], the resistance saturates for an increasing electric field (here at $R \approx 8.5\,k\Omega$), which unambiguously demonstrates the presence of zero-energy line modes[4,12,19]. In other words, although the perpendicular applied electric field induces a bandgap within the system[36], topologically-protected states at the K/K′ valleys persist, giving rise to helical valley transport (see the insets of Fig. 1d, e). The length-dependent conductance follows the Landau-Büttiker formula[14] $\sigma = \sigma_0 \left(1 + \frac{L}{\lambda_m}\right)^{-1}$, which yields a mean free path of $\lambda_m \approx 2.2\,\mu m$ with a channel length of $L = 0.7\,\mu m$ and the theoretical conductance of the domain wall of $\sigma_0 = 4\,e^2\,h^{-1}$ (where $e$ is the electronic charge and $h$ Planck's constant). With $\lambda_m > L$, ballistic charge transport supported by the domain wall is confirmed, highlighting the high quality of the device[8,14]. Worth to note, away from charge neutrality both devices show low resistance. In this regime, which is dominated by contact resistance, we expect no influence of the domain wall.

### Behaviour of the kink states in the presence of broken-symmetry phases at low magnetic field.

Whereas artificially constructed domain walls can only be investigated in the presence of a perpendicular electric field[4,7,8] in a limited range of electric fields and densities, quantum transport along stacking domain walls have mostly been studied in zero magnetic field[14]. Hence, we focus here on the interplay of topological domain walls and quantum Hall edge transport. Figure 2a, b shows the conductance in the devices D1-DW and D2 as a function of charge carrier density $n$ and electric field $E$ at a magnetic field of $B = 3$ T. In both devices, the broken-symmetry states within the lowest Landau level octet[24,28–31] appear, however, with very different conductance values (see Fig. 2c). The emerging quantum Hall states in device D1-DW, although exhibiting unusual conductance values, can unambiguously be identified by examining their slope in fan diagrams (see Supplementary Fig. 1). Thus, the stacking domain wall in device D1-DW contributes additional charge transport channels in parallel to the quantum Hall edge states altering the overall conductance of the device. In fact, tracking the conductance of both devices as a function of density (Fig. 2c) reveals a conductance offset for most of the appearing states. In device D1-DW, the $\nu = 0$ state at zero electric field, which has previously been identified as an insulating canted antiferromagnetic (CAF) state[37,38], shows a rather high conductance of $\sigma \approx 2.9\,e^2\,h^{-1}$ (see Fig. 2d). CAF states have been

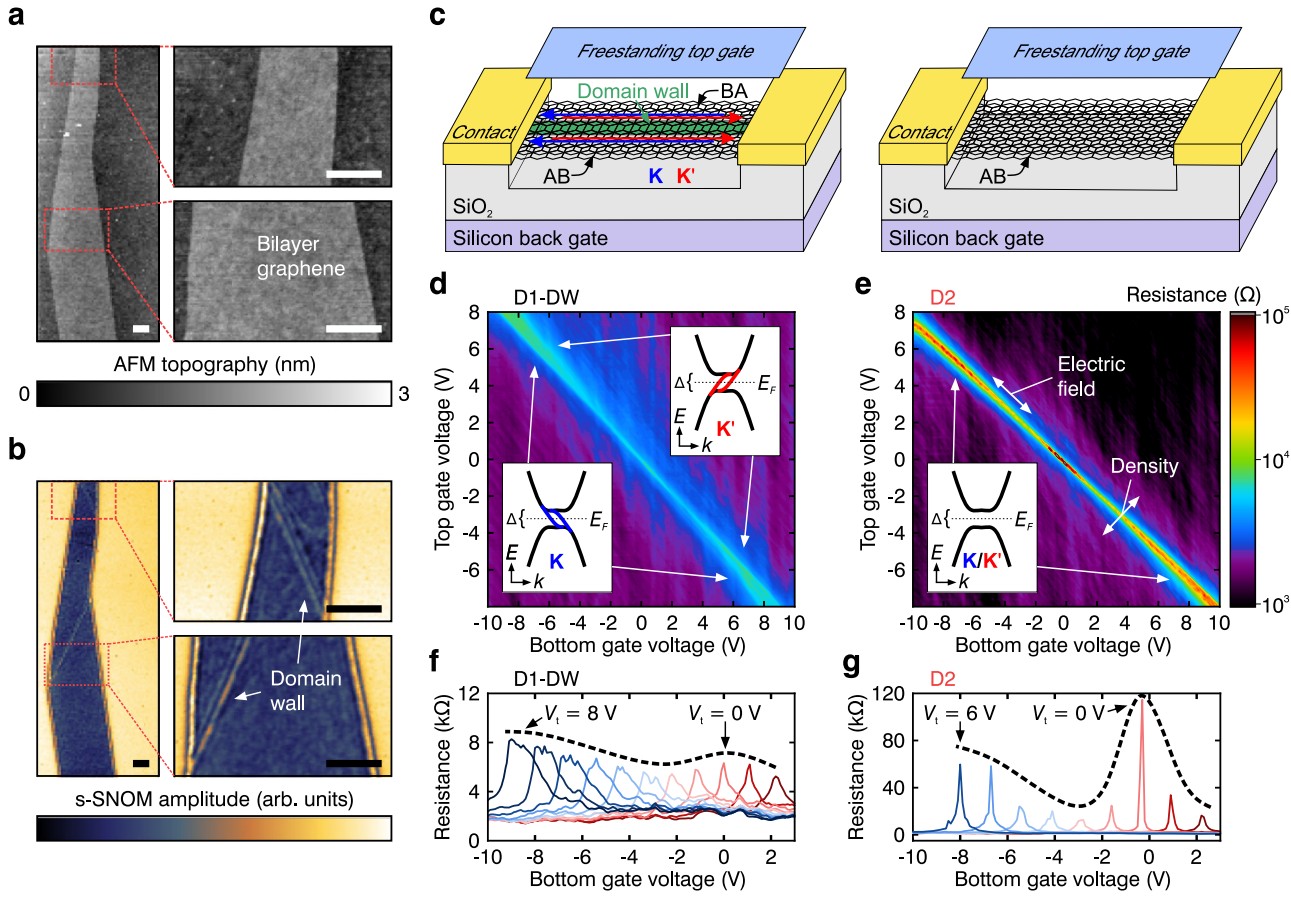

**Fig. 1 Topologically-protected states in bilayer graphene. a, b** Atomic force microscopy image (**a**) and scattering-type scanning near-field microscopy image (**b**) of a bilayer graphene flake, with high-resolution zoom-in scans on the right. The scale bars are 0.5 μm. **c** Freestanding dually gated bilayer graphene devices schematically shown with (left) and without domain wall (right) connecting the contacts. Topological valley transport along the domain wall is shown in blue and red in the K- and K'-valley, respectively. **d, e** Resistance map as a function of top and bottom gate voltage for device D1-DW (**d** with domain wall) and D2 (**e** without domain wall). Insets: Electronic band structure of bilayer graphene with (**d**) and without a domain wall (**e**) for an applied electric field. Δ is the electric field induced bandgap, $E_F$ the Fermi level and the blue (red) lines indicate topologically protected, doubly spin degenerate chiral states in the K(K')-valley. **f, g** Trace of the resistance as a function of $V_b$ for various $V_t$ with steps of 1 V shown for device D1-DW (**f**) and D2 (**g**). The dashed lines indicate the envelope of the resistance and are a guide to the eye.

observed to have low edge conductance, attributed to the opening of a spectral minigap at the sample edges[2,37,38]. The observed high conductance is thus consistent with the maximum possible —four—kink states at the DW contributing to the charge transport (with a finite $\lambda_m \approx 1.9\,\mu$m), as is also the case in the layer polarised (LP) $\nu = 0$ phase (see Supplementary Fig. 2 for more details) at high E. For an increasing filling factor, the conductance changes to $\sigma \approx 3.5, 4.0$ and $3.9\,e^2\,h^{-1}$ for the $\nu = -1, -2$, and $-4$ states (see Fig. 2d), respectively. This near constancy of conductance can be naturally explained: In the simplest model (see Fig. 2e), ignoring spin and orbital index for clarity, changing the Fermi level for an applied electrical field leads to the topological domain-wall channels being traded for quantum Hall edge channels. Changing the filling factor from the electron to the hole side, exchanges the positions of the valley polarised channels. More precisely (see Fig. 2f), when increasing the filling factor, a domain-wall channel disappears whereas an additional quantum Hall edge channel emerges. Hence, the conductance follows $\sigma = (4 - |\nu|)\sigma_{DW} + |\nu|\sigma_{QH}$ for $|\nu| \leq 4$, where $\sigma_{DW}$ is the conductance supported by a single kink state, and $\sigma_{QH} = e^2\,h^{-1}$ is the conductance of a quantum Hall edge channel. A linear fit to the data further supports this hypothesis (see Fig. 2d): for D2, it shows the expected slope of $1.0\,e^2\,h^{-1}$ per filling factor as there

are only quantum Hall edge states present. On the contrary, it yields a slope of $0.23\,e^2\,h^{-1}$ per filling factor for device D1-DW. Although in all $|\nu| \leq 4$ states four quantised channels contribute in total to the charge transport, the non-zero slope corresponds to the difference in conductance of the kink and edge states and shows that for increasing filling factor kink states with a conductance of $\sigma_{DW} \approx 0.77\,e^2\,h^{-1}$ are traded for higher-quality quantum Hall edge states with $\sigma_{QH} = e^2\,h^{-1}$. Discrepancies from the linear behaviour of the conductance in device D1-DW could indicate a distinct magnetic dependency of the conductance within the $|\nu| \leq 4$ states, as shown below. The $\nu = \pm 4$ states seem to be free of the influence of the domain wall (see Fig. 2f). A more detailed consideration of the band structure reveals that stacking domain walls can affect even the higher Landau levels, albeit more weakly (see Supplementary Fig. 3). In our freestanding devices, these states are at higher magnetic field outside the accessible density regime needed to observe the quantum Hall states.

**Emergence of a spectral minigap for high magnetic fields.** A more in-depth understanding of the intricate interplay between the quantum Hall edge modes and domain walls can be gained by investigating the charge transport at varying magnetic fields (see

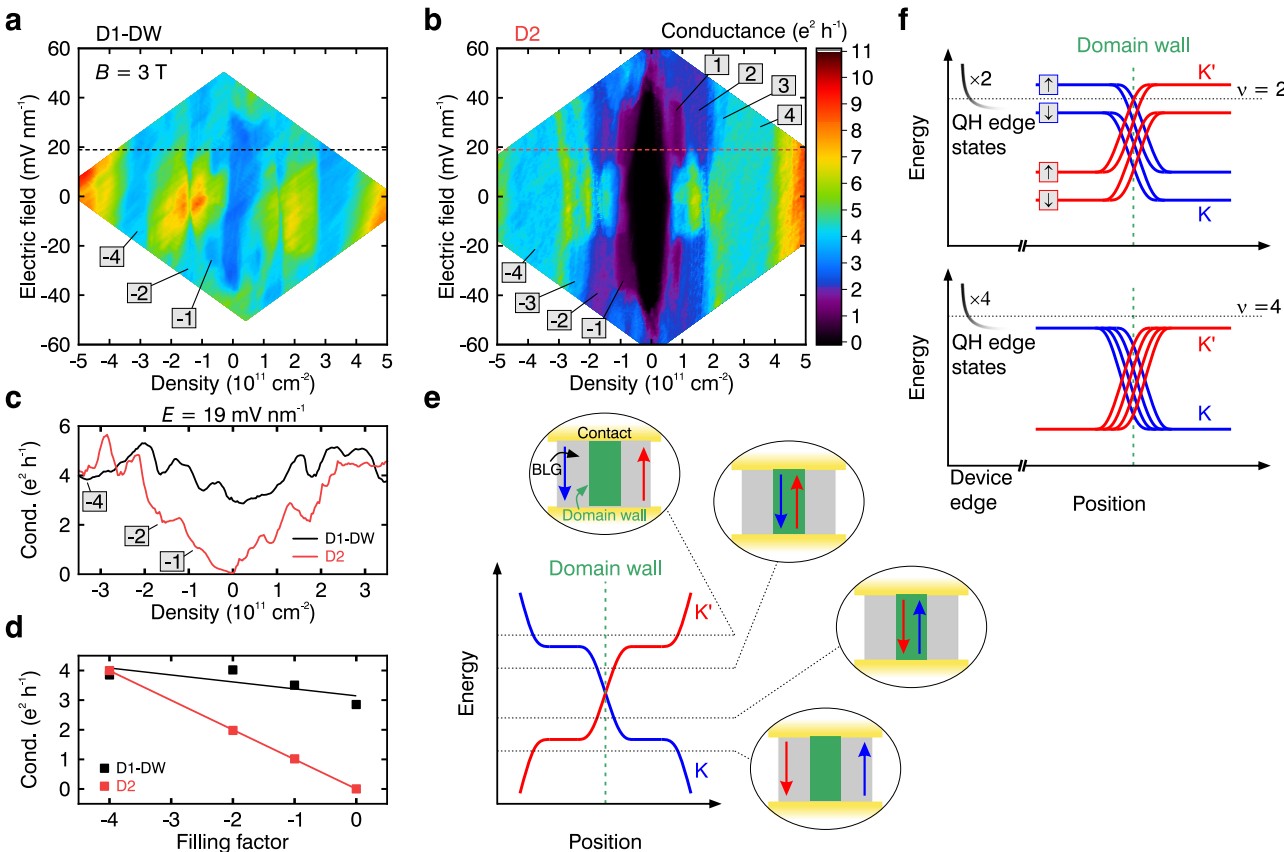

**Fig. 2 Interplay between topological valley and quantum Hall edge transport at low magnetic fields. a, b** Maps of the conductance in units of $e^2\,h^{-1}$ as a function of applied electric field $E$ and charge carrier density $n$ at a magnetic field of $B = 3$ T for devices D1-DW (**a**) and D2 (**b**). The dashed lines indicate the position of the data shown in **c**. Certain filling factors are indicated. **c** Line traces of the conductance as a function of $n$ taken at constant $E$ in device D1-DW (black) and D2 (red). **d** Conductance of quantum Hall states as a function of filling factor for device D1-DW (black) and D2 (red). The values are averaged over the electric field range at which the individual states emerge. The solid lines are linear fits to the corresponding data. **e** Schematic band structure (spin and orbital index omitted) in bilayer graphene in the presence of a stacking domain wall as a function of position. The dashed lines indicate distinct positions of the Fermi level and the corresponding encircled pictures schematically demonstrate the evolution of directions and locations of the one-dimensional channels within the device. **f** Schematic band structure as a function of position across the device with a domain wall shown for the $\nu = 2$ (top) and $\nu = 4$ (bottom) QH state in the presence of an interlayer electric field (spin and orbital flavours have been reinstated).

Fig. 3 and Supplementary Fig. 4 for more data). Line traces of the conductance as a function of filling factor measured in device D1-DW at zero and finite electric field show the $\nu = 0, \pm1, \pm2$ states (see Fig. 3a). In addition, we plot the conductance as a function of magnetic field for the individual states shown in Fig. 3b. Note that the conductance was averaged over the electric field range at which the respective state emerges, i.e. for the $\nu = 0$ CAF phase around zero electric fields, for the $\nu = -1$ and $-2$ at $|E| \geq 10\,\mathrm{mV\,nm}^{-1}$ and $|E| \geq 15\,\mathrm{mV\,nm}^{-1}$, respectively, and for the $\nu = -4$ state at all electric fields.

Most prominently, we see a sharp dip to very low conductance around zero charge carrier density within the $\nu = 0$ phase at high magnetic fields of $B \geq 8$ T (marked with a cross in Fig. 3a), which can also be tracked as function of magnetic field (see Fig. 3b). The feature is reproducible upon repeated sweeps and persists between different cooldowns of the device (see Supplementary Fig. 5). Towards $B = 0$, the $\nu = 0$ state corresponds to the layer antiferromagnetic phase with spin and valley indices locked[1,34,35]. In general, we find high conductance in this regime, suggesting the presence of zero-energy line modes at the kink. This observation would be consistent with the LAF order parameter experiencing an order parameter reversal as illustrated in Fig. 3c. The 1D modes persist within the gap because counterpropagating states in the same valley have opposite spin,

and hence scattering is suppressed. However, as the magnetic field is increased, spins cant and the LAF phase evolves into the canted antiferromagnetic phase[37,38]. Then, the counterpropagating modes in the same valley become partially spin aligned and can hybridise causing the emergence of a minigap. This is similar to the effect at the device edge. However, in the latter case the termination and backscattering off atomic scale defects can also couple opposite valleys[5], leading to further suppression of conductance. Our experimental data are indeed consistent with the opening of a gap and—when the Fermi level is located in this gap—a decrease in conductance. Outside of the gap, we expect a finite conductance, with a value determined by a sequence of the crossing bands and gap openings (see Fig. 3c). Since canting of spins gets stronger with magnetic field, one can expect the size of the minigap to grow with increasing $B$. This is consistent with our experimental observations of decreasing conductance (see Fig. 3a, b and Supplementary Fig. 4) and could be the reason why we can only resolve the minigap at $B \geq 8$ T. Eventually, for an infinite perpendicular or a finite in-plane magnetic field the CAF phase is expected to evolve into the ferromagnetic phase[37,38], in which the stacking domain wall has little or no effect on the Landau level energy (see Fig. 3c), making the stacking domain wall effectively invisible (this regime was not investigated experimentally in this study).

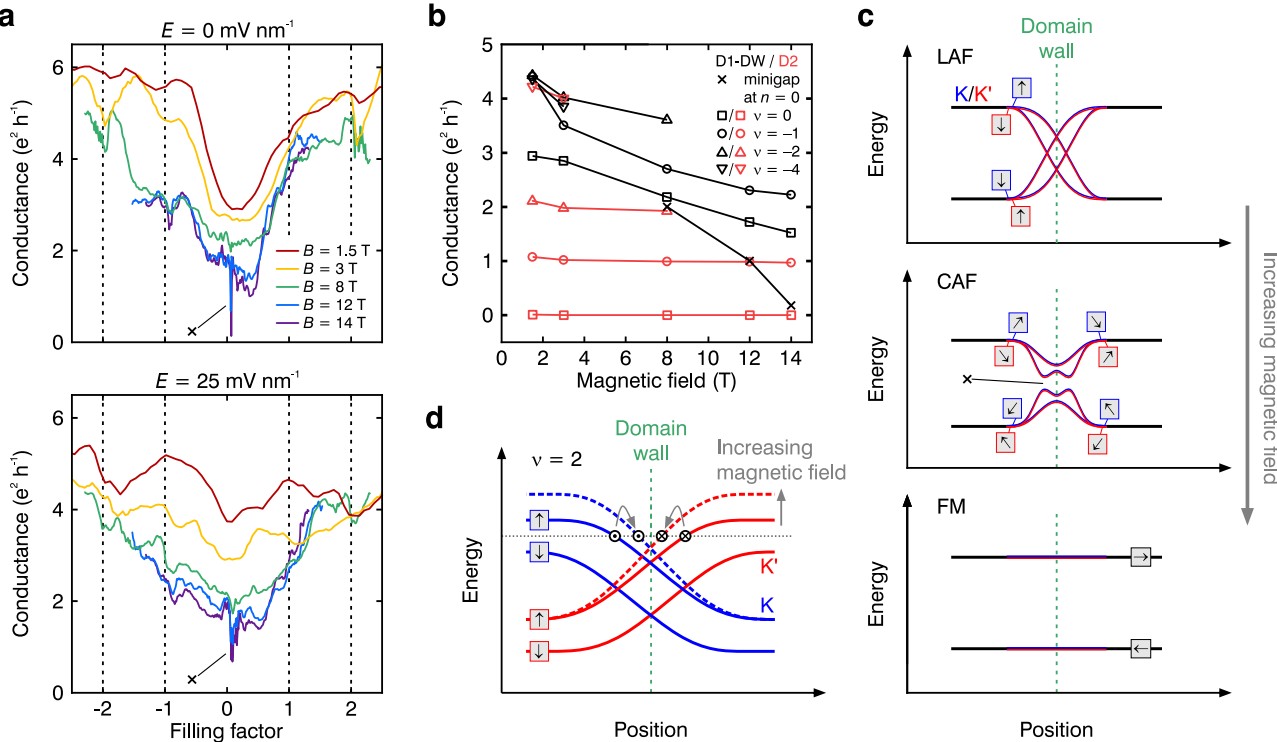

**Fig. 3 Behaviour of the kink states for varying magnetic field. a** Conductance as a function of filling factor shown for various magnetic fields at $E = 0$ (top) and $E = 25$ mV nm$^{-1}$ (bottom) measured in device D1-DW. The cross indicates the sharp conductance dip caused by the opening of a minigap. Note, that the state emerging around zero density is the LAF/CAF state, only at $E = 25$ mV nm$^{-1}$ the data curve for $B = 1.5$ T shows the transition between LAF/CAF and LP phase, see also Supplementary Fig. 2. **b** Conductance of the $\nu = 0, -1, -2, -4$ quantum Hall states as well as within the minigap as a function of magnetic field. The data for device D1-DW (D2) is shown in black (red). **c** Schematic band structure around the domain wall shown for the LAF, CAF and FM $\nu = 0$ phases. The blue (red) lines indicate the chiral states in the K(K')-valley. The cross indicates the spectral minigap emerging in the CAF phase. **d** Schematic band structure for $\nu = 2$ (orbital index is implicit) in the presence of layer-polarising bias. The domain wall retains only two pairs of valley helical (spin polarised) states, indicated by black circles with in-plane and out-of-plane directions. Their backscattering rate at the chemical potential (thin horizontal line) depends on their spatial separation and width. Both are generally expected to change as a function of magnetic field, leading to a change in DW conductance. The influence of the magnetic field is indicated by grey arrows. A similar effect was observed in artificial domain walls[7].

Notably, the conductance of the $\nu = \pm 1, \pm 2$ states also decrease with increasing $B$ (Fig. 3a, b), whereas device D2 shows the expected constant values as a function of $B$ for each QH state (see Fig. 3b). These quantum Hall states occur in sufficiently large electric field, and thus the valley polarisation is expected to change sign across the domain wall. In contrast, the spin polarisation remains constant across domain walls, pinned to the direction of magnetic field (see Fig. 2f). Therefore, the counter-propagating states at the domain wall belong to opposite valleys but same spin and can only be destroyed by local defects that can provide large momentum scattering. That is in contrast to the CAF state at $\nu = 0$ and $E = 0$, where a minigap can open owing to the hybridisation of states within the same valley and without the need for short range scattering. The measurements indicate that increasing the magnetic field increases the intervalley scattering, although the exact mechanism at this point remains unclear. One possible explanation could be the change in relative spatial arrangement of the counterpropagating channels as a function of magnetic field (see Fig. 3d). Clearly, increasing the channel separation should suppress backscattering, and vice versa. An effect of this type has already been observed at domain walls, where application of magnetic field or change of the chemical potential was found to affect the domain-wall conductance[7,39]. Another possibility could be the that increasing magnetic field pushes the system towards other broken-symmetry states[40,41], which would change the order parameter and hence

the behaviour of the kink states. However, these states have been observed only at very high magnetic fields and since we see no evidence of phase transitions in sample D2 for the same parameters, this possibility appears unlikely. Given that the measurements were performed in a two-terminal configuration, one should also make sure that the effect that we observe is not a consequence of a magnetic field dependent contact resistance of the kink states. However, we do not observe this behaviour for quantum Hall edge states (see Fig. 3b), and it is likely that the contact resistance of both types of one-dimensional channels behaves similarly. Additional devices revealed similar behaviours of the domain-wall conductance with increasing magnetic field (see Supplementary Fig. 6).

**Temperature dependence of the domain-wall states.** As final investigation to establish the interplay between edge and domain walls, we have conducted temperature dependent measurements. In Fig. 4, the conductance is shown as a function of temperature measured in different phases: in the layer antiferromagnetic, the canted antiferromagnetic as well as the layer polarised $\nu = 0$ phases and in the $\nu = -4$ phase. In contrast to device D2, which shows an activated temperature dependence of the conductance in all phases, D1-DW exhibits a much weaker temperature dependence and, most importantly, a finite conductance at low temperatures for the insulating LAF, CAF, and LP phases (see

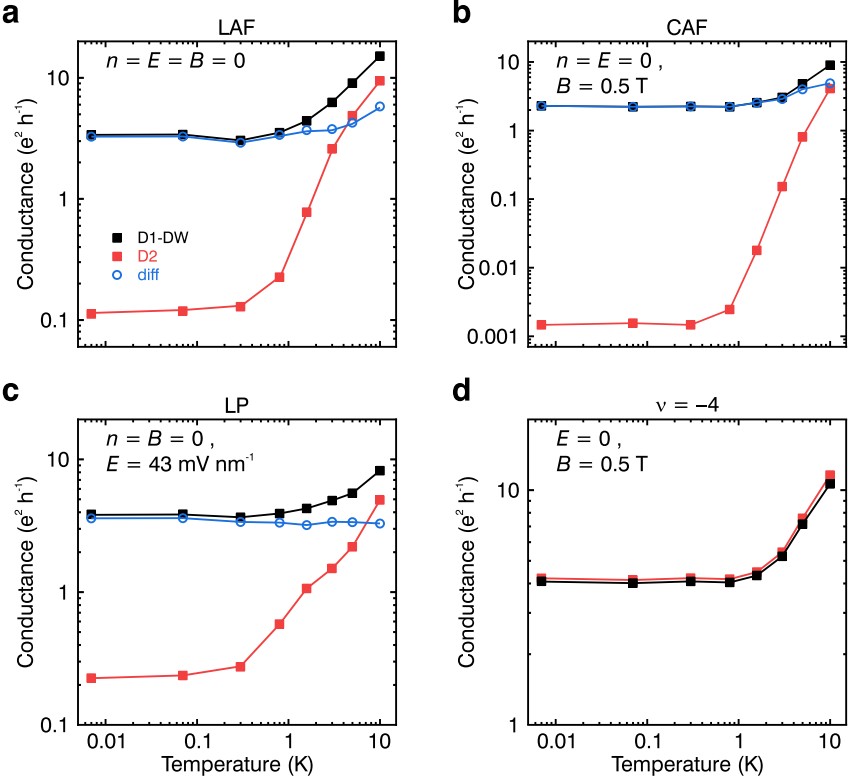

**Fig. 4 Temperature dependence of the conductance in various broken-symmetry phases. a–d** Temperature dependence of the conductance measured for the LAF phase at $n = E = B = 0$ (**a**), the CAF phase for $n = E = 0$ and $B = 0.5$ T (**b**), the LP phase at $n = B = 0$ and $E = 43$ mV nm$^{-1}$ (**c**) and the $\nu = -4$ phase at $E = 0$ and $B = 0.5$ T (**d**). The data corresponding to device D1-DW (D2) is shown in black (red). Moreover, in **a–c**, the difference of conductance between the two devices $\sigma_{diff}(T) = \sigma_{D1-DW}(T) - \sigma_{D2}(T)$ is shown as a function of temperature in blue. Note that the temperature dependence was measured in a different loading and annealing cycle than the measurements shown in Figs. 1–3 leading to small disparities in the conductance.

Fig. 4a–c, respectively). As the charge channels induced by the stacking domain wall contribute in parallel to any edge channels, we can subtract the data measured in both devices to reveal the underlying temperature dependence of the domain-wall $\sigma_{DW}(T) \approx \sigma_{diff}(T) = \sigma_{D1-DW}(T) - \sigma_{D2}(T)$, assuming that the activated charge transport behaves similarly in both devices. Notably, in all three $\nu = 0$ phases the difference $\sigma_{diff}(T)$ shows an approximately constant behaviour at low temperatures with $\sigma_{diff} \approx 2.5 - 3.5 \ e^2 \ h^{-1}$ and only a slight increase in the LAF and CAF phases for $T \geq 3$ K. Overall, this weak temperature dependence is expected for 1D charge transport and suggests weakly localised metallic behaviour[42]. On the contrary, the $\nu = -4$ phase (see Fig. 4d) shows the same activated temperature dependence and very similar conductance values in both devices, indicating that the domain wall has negligible influence on the quantum transport in this phase.

In conclusion, we have investigated the impact of stacking domain walls on the eightfold degenerate zero-energy Landau level in bilayer graphene. For future measurements, high in-plane magnetic fields would be beneficial to explore the behaviour of domain walls within the $\nu = 0$ ferromagnetic phase[38]. Moreover, the usage of encapsulated devices is essential to investigate the behaviour of domain walls in heterostructures[43] and their impact on the energy landscape of correlated states in higher Landau levels. Furthermore, having established that in the lowest Landau level the edge states and domain-wall channels co-exist, one can imagine investigating their mutual interaction[44] in narrow samples. Lastly, a direct imaging[39] of topological valley and quantum Hall edge channels would be very illuminating.

## Methods

Bilayer graphene was exfoliated from a highly ordered pyrolytic graphite (HOPG) block onto Si/SiO$_2$ substrates and suitable flakes were preselected using optical microscopy. Afterwards, infrared nano-imaging[45] was performed in a scattering-type scanning near-field microscope (s-SNOM, neaspec GmbH) in tapping mode to detect any stacking domain walls. Hereby, an infrared CO$_2$ laser beam (with a wavelength of 10.5 μm) was focused onto a metal-coated atomic force microscopy tips (Pt/Ir, Arrow NCPT-50, Nanoworld), which was oscillating with a frequency and amplitude of 250–270 kHz and 50–80 nm, respectively. With this method, we were able to obtain topographic and infrared nano-images simultaneously. Electrodes (Cr/Au, 5/100 nm) in two distinct configurations, a top gate (Cr/Au, 5/160 nm) as well as a spacer (SiO$_2$, 140 nm) were fabricated using several steps of standard lithography techniques and electron beam evaporation. Subsequently, the devices were submersed in hydrofluoric acid to etch about 150–200 nm of the SiO$_2$ and consequently suspend both the top gates and bilayer graphene flakes. After loading the freestanding dually gated bilayer graphene devices into a dilution refrigerator current annealing was performed at 1.6 K. In devices without domain wall best results were obtained when using a current of about 0.35 mA μm$^{-1}$ per layer. In devices with domain wall 150–250% more current was needed to achieve a current saturation due to their lower resistance and shorter channels. All quantum transport measurements were conducted at the base temperature of the cryostat ($T < 10$ mK), if not noted differently. Moreover, an excitation a.c. bias current of 0.1–10 nA at 78 Hz and Stanford Research Systems SR865A and SR830 lock-in amplifiers were used for the measurements, as well as Keithley 2450 SourceMeters to apply the gate voltages. Low-pass filters were used in series to reduce high frequency noise.

## Data availability

All data supporting the findings of this study are available within the article, as well as the Supplementary Information file, or available from the corresponding authors on request.

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

## Acknowledgements

R.T.W. and F.R.G. acknowledge funding from the Centre for Nanoscience (CeNS) and by the Deutsche Forschungsgemeinschaft (DFG, German Research Foundation) under Germany's Excellence Strategy-EXC-2111-390814868 (MCQST). I.M. was supported by the Materials Sciences and Engineering Division, Basic Energy Sciences, Office of Science, U.S. Dept. of Energy. We also thank Y.C. Durmaz and F. Keilmann for experimental assistance with the near-field optical microscopy.

## Author contributions

F.R.G. fabricated the devices and conducted the measurements and data analysis. I.M contributed the theoretical part. F.R.G., F.W, A.M.S, J.L, I.M., and R.T.W. discussed and interpreted the data. R.T.W. supervised the experiments and the analysis. The paper was prepared by F.R.G., I.M., and R.T.W with input from all authors.

## Funding

## Competing interests

The authors declare no competing interests.
