## [Peer Review File · Nature Communications]

REVIEWER COMMENTS

Reviewer #1 (Remarks to the Author):

This paper reports the interplay between topological domain wall states and quantum Hall edge transport in suspended bilayer graphene. The results are quite interesting and deserve to be published in Nature Communications. However, there are two issues that need to be addressed. I enter in more details in the following paragraphs.

First, as far as I can tell, the results are obtained in one sample with a DW. There is no indication of what happens at different lengths of the DW. I believe that the length of the DW is a critical variable here, according to Nature 520, 650–655 (2015), and the results from devices with different lengths are important to fully understand the phenomena reported in this work.

Second, the authors reported evidence of transport suppression at the DW in high magnetic fields and attributed this to the increase of the intervalley scattering. They discussed this phenomenon based on the variation in relative spatial distribution of the conducting channels along the DW in different magnetic fields. Actually, such a measurement has been studied in Nature Commun. 7, 11760 (2016) and I believe that it will be helpful to understand the high-field transport measurements.

Reviewer #2 (Remarks to the Author):

The manuscript entitled “Interplay between topological protected valley and quantum Hall edge transport” presented by Fabian R. Geisenhof et al., reported the study of the interplay between topological domain wall states and quantum Hall edge transport of high-quality suspended bilayer graphene by means of transport measurements. While the topologically protected states have been investigated before, their intriguing interplay remains poorly understood. This is an interesting field and the authors present a series of convincing data with an in-depth analysis and this innovative work is significant. I recommend it for publishing after some minor points are addressed.

1. Some of the figures in this manuscript is unclear. Such as Figure S1, the differential conductance fan diagrams are really blurry. The marked filling factors are quite hard to trace. A rather low-resolution image is hard to identify some of the details and less convincing. A possible solution is to adjust the image contrast or change another color scale to enhance the image quality. Similar problem occurs in a few other resistance maps. Reduce the color variety in the scale bar might help this situation.

2. The authors claimed that the dual-gate resistance map of mentioned two devices are not the same due to additional charge channels from device D1-DW, which mask the insulating phase. This statement agreed the spontaneously gapped state at the charge neutrality point and the insulating fully layer polarized state really well. But for the rest part of the dual-gate map, the background resistance of D1-DW is distinctly higher than D2 since the two figures share the same scale bar. It is hard to find a reasonable cause from this manuscript. Whether it is caused by contact resistance or the Domain wall resistance or something else. The authors should present an explanation for this phenomenon.

3. The authors claimed that “When increasing the filling factor, a domain wall channel disappears whereas an additional quantum hall edge channel emerges.” Figure 2d has a linear fit of the conductance data. However, the slope parameter seems to be imprecisely because the number of data point is not enough and unlikely to following the linear model. If there is another model could fit in this data better, it might make the conclusion in this manuscript more convincing.

4. Some figures in this manuscript are not strictly accordant in format. Especially in some schematic figures such as Figure S2e. A consistent format throughout the whole manuscript would make it looks better. Besides, there are some icons with inappropriate format in some of the figures. Quotation marks should print with halfwidth instead of fullwidth. For example, “K’ ” for “K’ ”.

5. Figure 3a shows the conductance of the kink states for varying magnetic fields and there is a sharp dip for higher magnetic field according to this image. Even this image was zoomed from the original data. The dip feature is still unclear due to a rather wide range of filling factor. This figure needs to zoom in more to get a more detailed dip feature.

Reviewer #3 (Remarks to the Author):

Geisenhof et al. reported magneto transport study of AB/BA stacking domain walls in suspended bilayer graphene devices. They compared devices with and without domains to set the stage, and then focused on the magnetic field-dependence of the domain wall transport at low carrier densities. They made comparisons with previous studies of AB/BA domain wall transport and artificially created field-induced domain walls.

Topological valley transport at bilayer graphene domain walls was established since 2015. This is the second transport study of stacking domain wall confirming the robust 1D transport of the domain wall channels. What’s new in this work are: 1. Presumably, the device quality should be higher than that of

devices where graphene is encapsulated by oxides (Ref. 14). So charge impurity scattering should be greatly suppressed; 2. The behavior of domain wall transport in a magnetic field. Comparing to the artificially field-induced domain transport, which was performed on hBN-encapsulated devices, this work shows almost quantized conductance in the absence of magnetic field. This fact poses a question on the role of hBN.

I have several questions/comments:

1. Have the authors performed transport in hBN-encapsulated bilayer graphene with domain walls? As I mentioned above, the role of hBN has been mysterious in the context of domain wall transport. In Ref. 7 & 8, the domain wall conductance was never quantized at the right value at < 4 Tesla (even for very short channels), which poses question on the exact nature of their observations. Naively, hBN encapsulation should suppress impurity scattering and result in cleaner transport than in Ref. 14. Can the authors at least comment on this?

2. Related to question 1, if the difference between this work and Ref.7&8 is that there are less impurities in the suspended device, why would the device resistance be so low in the field-induced gapped state? In Fig. 1g the peak resistance reaches only ~ 60 kOhm when the single-particle gap is opened. In contrast, (hBN-encapsulated) bilayer graphene device without domain wall can easily reach MOhm resistance.

3. I found the discussion of the back-scattering between valley-polarized channels very confusing at least. For example, at $\nu=0$, four states should be occupied. In the LAF state, both (K, spin up) and (K, spin down) should be occupied. However, in the Fig. 3c, the authors seem to imply that in the bulk there's a spin-valley locking?

4. Related to 3, I think the authors need to distinguish two pictures: the ground state of $D=n=0$, and the field-induced single-particle gapped states. Fig. 3a includes both $D=0$ and $D=a$ finite value, while the former is dominated by the manybody physics (where LAF is relevant) and the latter is a simple band insulator. Does $D=25$ mV/nm correspond to a band insulator?

5. Following 4, the way to see the existence and drawing the domain wall state is based on the simple band insulator band structure. If both $D=0$ and $D=25$ mV/nm correspond to the manybody state LAF, one cannot apply the band insulator band structure to the discussion of the topologically protected domain wall state/transport.

6. The expression is, unfortunately, very sloppy. Just to point out a few typos/wrong expressions:

a. The title used 'topological protected', while the whole abstract used 'topologically protected';

b. The word 'insulting' in Line 104 should be 'insulating';

c. The unit of magnetic field in Fig. S1 is wrong.

7. The authors claimed that they chose the suspended configuration is because hBN-encapsulation might get rid of the domain wall. This is wrong. People were able to see domain walls in hBN-encapsulated samples. See <https://pubs.acs.org/doi/abs/10.1021/acs.nanolett.1c00276>

8. To separate the orbital effect from the Zeeman effect, can the authors perform the measurement in a in-plane magnetic field? This can effectively separate the problem of Landau quantization and the spin-related backscattering problem, which is the theme of Figure. 3.

Overall, I found certain aspects of this work intriguing, but it needs to be improved significantly before can be published.

Referee #1 (Remarks to the Author):

This paper reports the interplay between topological domain wall states and quantum Hall edge transport in suspended bilayer graphene. The results are quite interesting and deserve to be published in Nature Communications.

Reply: We thank the referee for these encouraging comments about our work.

However, there are two issues that need to be addressed. I enter in more details in the following paragraphs.

1. First, as far as I can tell, the results are obtained in one sample with a DW. There is no indication of what happens at different lengths of the DW. I believe that the length of the DW is a critical variable here, according to Nature 520, 650–655 (2015), and the results from devices with different lengths are important to fully understand the phenomena reported in this work.

Reply: We thank the referee for this comment. First, we would like to point out that we have investigated more than one sample containing a domain wall (see Fig. S6 for data of further devices). And second, we certainly agree with the referee that the length of the channel plays an important role for the conductance. In the cited ref. 14 (Nature 520, 650–655 (2015)), as the referee pointed out, as well as in ref. 7 there are measurements shown which determine the length-dependency of the domain walls conductance. Hence, we have not focused on this already explored dependency. However, on page 5, where we calculate the domain walls conductance following the Landau-Büttiker formula, we have used the channel width. We agree with the referee, that this should be pointed out more clearly and have changed the paragraph on page 5 as follows:

“The length-dependent conductance follows the Landau-Büttiker formula¹⁴ $\sigma = \sigma_0 \left(1 + \frac{L}{\lambda_m}\right)^{-1}$, which yields a mean free path of $\lambda_m \approx 2.2 \mu\text{m}$ with a channel length of $L = 0.7 \mu\text{m}$ and the theoretical conductance of the domain wall of $\sigma_0 = 4 e^2 h^{-1}$ (where e is the electronic charge and h Planck’s constant).”

2. Second, the authors reported evidence of transport suppression at the DW in high magnetic fields and attributed this to the increase of the intervalley scattering. They discussed this phenomenon based on the variation in relative spatial distribution of the conducting channels along the DW in different magnetic fields. Actually, such a measurement has been studied in Nature Commun. 7, 11760 (2016) and I believe that it will be helpful to understand the high-field transport measurements.

Reply: We would like to thank the referee for this comment and for pointing out the interesting study where the authors can directly image the topological edge states at the domain walls using STM. Furthermore, measurements at high magnetic fields are shown. For the behaviour of the edge states in high magnetic fields a similar observation to Ref. 14 (10.1038/NNANO.2016.158) is given, namely that by increasing the magnetic field the edge states move further spatially apart. In our case, at low electric fields, we a similar effect, i.e. the magnetic field is moving the states further apart (see Fig. 3d). We agree that discussing

the given reference by the referee is certainly helpful and improves the discussion about the behaviour in high magnetic fields. Therefore, we have added the reference to the corresponding paragraph on page 9:

“An effect of this type has already been observed at domain walls, where application of magnetic field or change of the chemical potential was found to affect the domain wall conductance^{7,39}.”

Here, Ref. 39 is the newly added reference suggested by the referee.

Moreover, we truly find the direct imaging of the edge states very intriguing and want to highlight it as possibility for future measurements. With the help of this method, the topological valley and quantum Hall edge states could directly be distinguished. We have added a sentence in to the outlook, mentioning this in the manuscript, see page 11:

“Lastly, a direct imaging³⁹ of topological valley and quantum Hall edge channels would be very illuminating.”

Referee #2 (Remarks to the Author):

The manuscript entitled “Interplay between topological protected valley and quantum Hall edge transport” presented by Fabian R. Geisenhof et al., reported the study of the interplay between topological domain wall states and quantum Hall edge transport of high-quality suspended bilayer graphene by means of transport measurements. While the topologically protected states have been investigated before, their intriguing interplay remains poorly understood. This is an interesting field and the authors present a series of convincing data with an in-depth analysis and this innovative work is significant. I recommend it for publishing after some minor points are addressed.

Reply: We thank the referee for the appreciation of our work. We have addressed all comments below and we think the effort has significantly improved the work.

1. Some of the figures in this manuscript is unclear. Such as Figure S1, the differential conductance fan diagrams are really blurry. The marked filling factors are quite hard to trace. A rather low-resolution image is hard to identify some of the details and less convincing. A possible solution is to adjust the image contrast or change another color scale to enhance the image quality. Similar problem occurs in a few other resistance maps. Reduce the color variety in the scale bar might help this situation.

Reply: We thank the referee for this comment. During the submission process, we have used images with rather limited resolution (only 300 dpi). However, for the final version we could provide high-quality vector graphics, which will certainly help for higher image quality.

2. The authors claimed that the dual-gate resistance map of mentioned two devices are not the same due to additional charge channels from device D1-DW, which mask the insulating phase. This statement agreed the spontaneously gapped state at the charge neutrality point and the insulating fully layer polarized state really well. But for the rest part of the dual-gate map, the background resistance of D1-DW is distinctly higher than D2 since the two figures share the same scale bar. It is hard to find a reasonable cause from this manuscript. Whether

it is caused by contact resistance or the Domain wall resistance or something else. The authors should present an explanation for this phenomenon.

Reply: We thank the referee for this comment and agree that the discussion about the behaviour at non-zero charge carrier density for zero magnetic field is missing. As the referee pointed out, we have discussed the additional charge channels from device D1-DW in the fully layer polarized state, i.e. at $n=B=0$ but non-zero electric field, and also for the insulating layer antiferromagnetic state, i.e. at $n=B=E=0$. In fact, from the band structure (see inset Fig.1d) we expect the domain walls only contribute to the charge transport in case the bulk of the bilayer is gapped, which are for example the two above mentioned cases. Outside of the gap, i.e. for positive or negative charge carrier density we do not expect any influence of the domain walls. The differences at high charge carrier density in the resistance map of device D1-DW and D2 are rather originating from contact resistance, as the referee already suggested. To include this important information in the manuscript, we have added the following sentence on page 5:

“Worth to note, away from charge neutrality both devices show low resistance. In this regime, which is dominated by contact resistance, we expect no influence of the domain wall.”

3. The authors claimed that “When increasing the filling factor, a domain wall channel disappears whereas an additional quantum hall edge channel emerges.” Figure 2d has a linear fit of the conductance data. However, the slope parameter seems to be imprecisely because the number of data point is not enough and unlikely to following the linear model. If there is another model could fit in this data better, it might make the conclusion in this manuscript more convincible.

Reply: We thank the referee for this comment. We agree that conductance of device D1-DW of the $|\nu| \leq 4$ states show not a clear linear behaviour. From our theoretical considerations we expect four states in all of the $|\nu| \leq 4$ states, hence, without any backscattering and consequently perfect kink and edge states, we would see a constant conductance of $4 e^2 h^{-1}$ within the $|\nu| \leq 4$ states. Still, the conductance would come from different states, as for example in the $\nu = -2$ state two kink states are present (and two quantum Hall edge channels), whereas in the $\nu = 0$ state four kink states are present (and zero quantum Hall edge channels). However, since for increasing filling factor single kink states are traded for quantum Hall edge channels, and our data suggests that the former have lower quality than the latter, we expect a linear rather than a constant behaviour of the conductance for varying filling factor within the $|\nu| \leq 4$ states. We agree that our data shows discrepancies from the linear model. These discrepancies could easily come from different contact resistance of individual kink states. Another possibility could be, that the intervalley backscattering we see for increasing magnetic field can be different for the individual quantum Hall states, see Fig. 3b. Still, we want to keep the model as simple as possible and think it describes the observations well enough. Nonetheless, we agree with the referee, that the uncertainty should be commented on in the manuscript. To this end, we have edited the paragraph on page 7 as follows:

“Although in all $|\nu| \leq 4$ states four quantized channels contribute in total to the charge transport, the non-zero slope corresponds to the difference in conductance of the kink and edge states and shows that for increasing filling factor kink states with a conductance of $\sigma_{DW} \approx 0.77 e^2 h^{-1}$ are traded for higher-quality quantum Hall edge states with $\sigma_{QH} = e^2 h^{-1}$. Discrepancies from the linear behaviour of the conductance in device D1-DW could

indicate a distinct magnetic dependency of the conductance within the $|v| \leq 4$ states, as shown below.”

4. Some figures in this manuscript are not strictly accordant in format. Especially in some schematic figures such as Figure S2e. A consistent format throughout the whole manuscript would make it look better. Besides, there are some icons with inappropriate format in some of the figures. Quotation marks should print with halfwidth instead of fullwidth. For example, “K' ” for “K’ ”.

Reply: We want to thank the referee for pointing out the suggestions regarding the format. To address these concerns, we have improved the colours of Fig. 1c and also Fig. 2e, so that the schematic figures are more consistent with the format of the other panels. We find this improved the consistency significantly. If the referee has any more suggestions regarding the format of the figures, we would be happy to implement them to further improve the quality of figures. Moreover, we have changed the prime mark as the referee suggested in the entire manuscript, and especially in the figures.

In addition, we have taken the opportunity to change the format of the units of conductance, electric field, etc. to be conform with the maths formatting of Nature Communications, i.e. changed mV/nm to mV nm⁻¹, etc. in the text as well as Figures. We have highlighted all changes in yellow.

5. Figure 3a shows the conductance of the kink states for varying magnetic fields and there is a sharp dip for higher magnetic field according to this image. Even this image was zoomed from the original data. The dip feature is still unclear due to a rather wide range of filling factor. This figure needs to zoom in more to get a more detailed dip feature.

Reply: We would like to thank the referee for this comment. We agree that the data on the sharp dip for high magnetic fields is rather limited. We would like to point out, that the feature is reproducible upon repeated sweeps, and importantly persists for different cooldowns of the device. To emphasize this robust observation, we have added a new Supplementary Figure S5 (the old Fig. S5 is now named Fig. S6). The figure shows a fan diagram recorded for up to B=14 T, where the minigap is appearing for high magnetic fields. Moreover, several back gate voltage sweeps show the minigap in detail. The data shows that the feature is persistent even for different cooldowns. Notably, during this cooldown, the device was not as clean as during the measurements shown in the main manuscript.

In the main manuscript, we have added a sentence on page 8 referring to the newly added Fig. S5 and the corresponding paragraph:

“The feature is reproducible upon repeated sweeps and persists between different cooldowns of the device (see Supplementary Fig. S5).”

The newly added Supplementary Fig. S5 and the corresponding text section is as follows:

“Persistence of the spectral minigap for a different cooldown

Fig. S5a shows a fan diagram recorded as a function of back gate voltage in device D1-DW. The graph shows the emergence of the spectral minigap within the $\nu = 0$ phase at $B \geq 8$ T, indicated by the cross. Moreover, its evolution for increasing magnetic field can be seen in Fig. S5b, which shows line traces of the conductance for various high magnetic fields. The dip in conductance is increasing for increasing magnetic field, which matches the observations shown in Fig. 3 in the main manuscript.

It is worth noting that the data shown in Fig. S5 was recorded during a different cooldown of the device than the data shown in the main manuscript. Most importantly, this demonstrates the persistence of the spectral minigap over multiple cooldowns. Since each cooldown involves a current annealing procedure, driving high currents through the device seem also to not affect the emergence of the feature. Notably, the device D1-DW was not as clean during the cooldown corresponding to Fig. S5 as for the measurements shown in the main manuscript (in terms of residual charge disorder and contact resistance), which makes the direct comparison of absolute values of the conductance difficult.

Fig. S5 | The emergence of a spectral minigap within the $\nu = 0$ phase for high magnetic fields. a, Fan diagram showing the conductance as a function of magnetic field and bottom gate voltage. The cross indicates the conductance dip caused by the emergence of the spectral minigap. Note that the data was recorded with device D1-DW but during a different cooldown than the measurements shown in the main manuscript. **b**, Line traces of the conductance as a function of back gate voltage at various magnetic fields. The data is taken from the fan diagram shown in (a). The line cuts are offset for better visibility.”

Referee #3 (Remarks to the Author):

Geisenhof et al. reported magneto transport study of AB/BA stacking domain walls in suspended bilayer graphene devices. They compared devices with and without domains to set the stage, and then focused on the magnetic field-dependence of the domain wall transport at low carrier densities. They made comparisons with previous studies of AB/BA domain wall transport and artificially created field-induced domain walls.

Topological valley transport at bilayer graphene domain walls was established since 2015. This is the second transport study of stacking domain wall confirming the robust 1D transport of the domain wall channels. What’s new in this work are: 1. Presumably, the device quality should be higher than that of devices where graphene is encapsulated by oxides (Ref. 14). So charge impurity scattering should be greatly suppressed; 2. The behavior

of domain wall transport in a magnetic field. Comparing to the artificially field-induced domain transport, which was performed on hBN-encapsulated devices, this work shows almost quantized conductance in the absence of magnetic field. This fact poses a question on the role of hBN.

I have several questions/comments:

1. Have the authors performed transport in hBN-encapsulated bilayer graphene with domain walls? As I mentioned above, the role of hBN has been mysterious in the context of domain wall transport. In Ref. 7 & 8, the domain wall conductance was never quantized at the right value at < 4 Tesla (even for very short channels), which poses question on the exact nature of their observations. Naively, hBN encapsulation should suppress impurity scattering and result in cleaner transport than in Ref. 14. Can the authors at least comment on this?

Reply: We thank the referee for this comment. In the course of this work, we have only investigated suspended devices. We agree with the referee, that the role of hBN is rather mysterious in the context of domain wall transport. When comparing our measurements to the one earlier reported in literature, two things need to be considered:

For one, we agree that devices encapsulated in hBN show certainly reduced impurity scattering compared to conventional graphene devices on silicon dioxide. However, suspended devices show also very high quality, as has been shown for example in Solid State Communications 146 (2008) 351–355. Moreover, the presence of hBN does suppress interaction effects and hence prevents the emergence of certain interaction induced phases near charge neutrality such as the layer antiferromagnetic phase. To see the impact of domain walls on the pure bilayer graphene system, we have therefore chosen to investigate suspended devices.

On the other hand, references [7] & [8] investigate hBN encapsulated devices but with artificially created domain walls rather than stacking domain walls. Although behaving similarly in certain aspects, the two types have important differences. Most prominently, stacking domain walls do not require application of electric field. Moreover, one could assume that artificial domain walls are more extended spatially (due to a non-zero gap between the split gates, imperfect edges of the gate electrodes, etc.) while the stacking domain walls are confined on a few nanometres. Possibly, this could lead to increased backscattering mechanisms.

The two aspects explained above give reasons for the exceptional quality of our devices and the investigated domain walls, which reveals itself in the long mean free path. Moreover, it explains why we see suspended devices as a convenient device architecture. Nonetheless, we certainly agree with the referee that role of stacking domain walls in encapsulated devices is mysterious and support the idea of this being a deserving subject for future studies. To express this also in our manuscript, we have included a sentence in the outlook on page 11 reading:

“Moreover, the usage of encapsulated devices is essential to investigate the behaviour of domain walls in heterostructures⁴³ and their impact on the energy landscape of higher Landau levels.”

2. Related to question 1, if the difference between this work and Ref.7&8 is that there are less impurities in the suspended device, why would the device resistance be so low in the field-induced gapped state? In Fig. 1g the peak resistance reaches only ~ 60 kOhm when the

single-particle gap is opened. In contrast, (hBN-encapsulated) bilayer graphene device without domain wall can easily reach MOhm resistance.

Reply: We thank the referee for this expert question. Indeed, the resistance in the electric field induced fully layer polarized $\nu=0$ phase in the device without domain wall (D2) reaches only $\sim 60\text{k}\Omega$, which seems rather low. The reason for this is the rather low electric field applied. For the gate voltage of $V_{\text{bottom}} = -8$ and $V_{\text{top}} = 6\text{V}$, as shown in Fig. 1g, the electric field is roughly $\sim 50\text{ mV/nm}$, which is rather low in comparison to the electric fields applicable for example in hBN encapsulated devices (in the order of up to 1 V/nm , see for example Science **375**, 6582, 774-778). In contrast to encapsulated devices, suspended devices are rather limited in the possible gate voltages. For such low electric fields, we do not expect very high resistances in our suspended devices for the fully layer polarised phase, since similar electric field ranges applied in comparable devices show similar results, see for example Nature Nanotech **7**, 156–160 (2012) and Nature **598**, 53–58 (2021). As a consequence, we cannot reach such high electric field, where the resistance reaches easily MOhms.

3. I found the discussion of the back-scattering between valley-polarized channels very confusing at least. For example, at $\nu=0$, four states should be occupied. In the LAF state, both (K, spin up) and (K, spin down) should be occupied. However, in the Fig. 3c, the authors seem to imply that in the bulk there's a spin-valley locking?

Reply: We thank the referee for this expert question. Indeed, the $\nu=0$ LAF state shows spin-valley locking (in contrast to the fully layer polarized $\nu=0$ state at high electric fields). This has been mentioned in other studies, see for example Nature Phys. **9**, 154–158 (2013) and Phys. Rev. B **86**, 075450 (2012).

4. Related to 3, I think the authors need to distinguish two pictures: the ground state of $D=n=0$, and the field-induced single-particle gapped states. Fig. 3a includes both $D=0$ and $D=a$ finite value, while the former is dominated by the manybody physics (where LAF is relevant) and the latter is a simple band insulator. Does $D=25\text{ mV/nm}$ correspond to a band insulator?

Reply: We thank the referee for raising this important point. We agree with the referee that it is very important to distinguish between the $\nu=0$ state around zero electric field (the LAF/CAF state) and the electric field induced fully layer polarised $\nu=0$ state at high electric fields (LP state). Throughout the manuscript, we have mainly focused on the low electric field regime, since it is mostly unexplored in previous studies. Nonetheless, in the Supplementary Fig. S2 we compare the two different $\nu=0$ phases for low and high electric fields.

In Fig. 3a, we show linecuts of the conductance as a function of filling factor at various magnetic fields for $E = 0$. We have included the data also for $E = 25\text{ mV/nm}$, since some states as the layer polarised $\nu=+1, -2$ are more easily visible for finite electric fields. The $\nu=0$ state is in both cases the LAF/CAF state, however, the linecut show for $B=1.5\text{ T}$ and $E = 25\text{ mV/nm}$ actually shows the transition region between LAF/CAF and LP phase (this fact can be seen in the Supplementary Fig. S2a). We thank the referee for pointing this out, and we have added a note to the caption of Fig. 3 as follows:

Note, that the state emerging around zero density is the LAF/CAF state, only at $E = 25 \text{ mV nm}^{-1}$ the data curve for $B = 1.5 \text{ T}$ shows the transition between LAF/CAF and LP phase, see also Supplementary Fig. S2.

5. Following 4, the way to see the existence and drawing the domain wall state is based on the simple band insulator band structure. If both $D=0$ and $D=25 \text{ mV/nm}$ correspond to the manybody state LAF, one cannot apply the band insulator band structure to the discussion of the topologically protected domain wall state/transport.

Reply: We thank the referee for this comment. The proper description of states with spontaneously broken symmetry, such as LAF, is indeed a nontrivial matter. In our analysis we adopted the mean-field description of the LAF/CAF phase (see for instance Fig 3C). Such description has proven to be both accurate and insightful. For instance, such mean field level diagram has been used to predict the sequence of transitions in BLG in the absence of domain walls, see for example Fig. 3a in Nat. Commun 8, 948 (2017). Even though to our knowledge this approach had not been applied to the problem of domain walls, it has been used extensively to analyse the closely related problem of edge physics; see for example Nature Phys. 9, 154–158 (2013), Phys. Rev. B 86, 075450 (2012) and Phys. Rev. B 86, 195435 (2012), where similar construction was used at $B = 0$ to argue that FM state has to have metallic edge states, while other states, such as LP and CAF should have none. Such arguments rely on a combination of band structure continuity and examining the possibility of mixing (scattering) between counterpropagating edge or domain wall states.

Nevertheless, the referee is correct that for a state with spontaneously broken symmetry, in the presence of a domain wall there may be an ambiguity about the OP value/direction on the two sides of the junction, and this can have a major effect on the conduction of the domain wall, particularly at $n = 0$.

These questions are discussed in detail in section *“Emergence of a spectral minigap for high magnetic fields.”* We want to emphasize that we already stated in the abstract that *“For high magnetic fields, however, we observe evidence of transport suppression at the domain wall, which can be attributed to the emergence of spectral minigaps. This indicates that stacking domain walls potentially do not correspond to a topological domain wall in the order parameter.”*

6. The expression is, unfortunately, very sloppy. Just to point out a few typos/wrong expressions:

- a. The title used ‘topological protected’, while the whole abstract used ‘topologically protected’;**
- b. The word ‘insulting’ in Line 104 should be ‘insulating’;**
- c. The unit of magnetic field in Fig. S1 is wrong.**

Reply: We thank the referee for pointing out the wrong expressions. We have gone through the manuscript again to make sure the expressions ‘topological’ and ‘topologically’ are correctly used. In this sense, we would like to change the title from “Interplay between topological protected valley and quantum Hall edge transport” to the correct and simpler version: *“Interplay between topological valley and quantum Hall edge transport”*. We hope the referee agrees with the change.

Furthermore, we have corrected the error on line 104 and corrected the unit of the magnetic field in Fig. S1. Moreover, we want to point out that we have changed the format of the units

of conductance, electric field, etc. to be consistent with the maths formatting of Nature Communications, i.e. changed mV/nm to mV nm⁻¹, etc. in the text as well as Figures. All changes are highlighted in yellow.

7. The authors claimed that they chose the suspended configuration is because hBN-encapsulation might get rid of the domain wall. This is wrong. People were able to see domain walls in hBN-encapsulated samples. See <https://pubs.acs.org/doi/abs/10.1021/acs.nanolett.1c00276>

Reply: We thank the referee for this comment. The cited references “Nanoimaging of Low-Loss Plasmonic Waveguide Modes in a Graphene Nanoribbon” is about monolayer graphene and shows the presence of plasmons in a graphene nanoribbon encapsulated in hBN. However, it does not show any domain walls and we think it is not fitting for this matter. Nonetheless, we certainly agree with the referee, that domain walls in general have been observed in hBN encapsulated samples (see for example Nano Lett. 2021, 21, 1688–1693). Perhaps the reviewer could also point us to additional references that we might have overlooked?

To address the referee’s valid comment, we adapted our motivation for choosing suspended devices in the introductory part of the manuscript on page 4 as follows:

“We chose freestanding dually gated bilayer graphene devices as an ideal and versatile platform, since on the one side – as indicated by our measurements below – DWs remain stable during processing and suspension, and, on the other side, suspending enables the investigation of quantum transport unaffected by surroundings. “

Moreover, as in the referee’s comment 1, since the role of hBN is indeed mysterious for domain walls transport we agree that it could be subject of future studies. To highlight this in our manuscript, we have added a sentence about the role of hBN in the outlook on page 11 and added the corresponding reference to give the observation of domain walls in hBN samples more credit:

“Moreover, the usage of encapsulated devices is essential to investigate the behaviour of domain walls in heterostructures⁴³ and their impact on the energy landscape of correlated states in higher Landau levels.”

8. To separate the orbital effect from the Zeeman effect, can the authors perform the measurement in a in-plane magnetic field? This can effectively separate the problem of Landau quantization and the spin-related backscattering problem, which is the theme of Figure. 3.

Reply: We thank the referee for this comment. We certainly agree with the referee that this would be a very intriguing measurement. In particular, for the observation of the $\nu=0$ ferromagnetic phase an in-plane magnetic field would be highly beneficial. Unfortunately, we think this is beyond the scope of the manuscript. For now, we are not able to perform these in-plane measurements. Nonetheless, we want to highlight the interesting idea in the manuscript to hopefully stimulate future measurements with in-plane fields. We have already mentioned this in the original outlook, but we have rewritten the sentence to highlight the idea even more, see page 11:

“For future measurements, high in-plane magnetic fields would be beneficial to explore the behaviour of domain walls within the $v=0$ ferromagnetic phase³⁸.”

Overall, I found certain aspects of this work intriguing, but it needs to be improved significantly before can be published.

Reply: We thank the referee for the encouraging feedback on our work. We have made efforts to fully address all the questions and comments above, which we find significantly improved our manuscript.

REVIEWERS' COMMENTS

Reviewer #1 (Remarks to the Author):

The authors have addressed with some thoroughness the points I raised, and I am now ready to recommend publication.

Reviewer #2 (Remarks to the Author):

All my questions have been addressed properly. Thanks. I recommend it for publishing.

Reviewer #3 (Remarks to the Author):

I'm fine with the revised manuscript being published at Nature Comm.